# A multi-dimensional analysis of CBEC English genre variation in South Asia: Based on Daraz

Shuang Wang 📇 *

Business English Department, Foreign Language College, Fujian Jiangxia University, Fuzhou, Fujian, China

* 190659117@qq.com

**Data Availability Statement:** All relevant data are within the paper and its Supporting information files.

**Funding:** SW received funding from the Center for Sri Lankan Studies, Foreign Language College, Fujianjiangxia University. The funder played a role

## Abstract

The purpose of this study is to investigate how genres vary and figure out the factors that generate genre variation. The quantitative multi-dimensional analysis is used to examine genre variation of cross-border e-commerce English in South Asia. The texts in the observed corpus collected from four country websites of Daraz, a significant cross-border e-commerce platform, were tagged and analyzed using Multidimensional Analysis Tagger and SPSS statistical software. The results of linear regression analysis, independent sample t-test and Analysis of Variance show salient differences between the observed corpus and reference corpus. The research also indicates that four sub-corpora from country Websites are brought into line with each other. They show salient differences (p<0.05) with the general corpus in two dimensions of linguistic variation. The findings indicate that variables that lead to the cross-border e-commerce English genre variation might be attributable to cultural backgrounds, regional market sizes, and fundamental internet facilities. In conclusion, these findings lend significant empirical support to systemic functional theories and suggest further research in the application of a multi-dimensional analysis approach in the cross-border e-commerce English genre.

## Introduction

With the change of business development direction of cross-border e-commerce (CBEC) enterprises, from "product going to sea" to "brand going to sea", it is a tendency that more cross-border e-commerce enterprises rely on customer service to build up a reliable brand image in the global markets, especially the online replies to customer inquiries and review. CBEC genre has drawn the observers' attention to genre variation research since this industry became booming worldwide. Some scholars believe that globalization, at least in the Anglo-Saxon world, allows genres to cross borders [1, 2]. Xu and Lockwood suggested that for a genre that depends extensively on specific business needs, companies can conduct genre analyzes on their personal webchat data for training materials that fit their context best [3].

The CBEC business in Daraz covers four of the seven countries in South Asia, including Nepal, Pakistan, Bangladesh, Sri Lanka, and one Southeast country, Myanmar. For geographical consideration, the raw corpus materials were collected from four other Daraz country websites in South Asia. According to incomplete statistics from the World Bank, from 2020–2021,

in decision to publish. URLs to sponsors' websites: https://fld.fjjxu.edu.cn/2415/list.htm.

**Competing interests:** The author has declared that no competing interests exist.

the four countries with a population of 429 million reached a total GDP of US$744.32 billion. The prospect of the CBEC blue ocean in South Asia has attracted many Chinese enterprises, so this paper takes Daraz, a local cross-border e-commerce platform, as the research object of genre variation, where the online replies to customer inquiries labeled as *answered questions* were collected as the observed corpus. Within the framework of the grammar system itself, Chinese brands in cross-border e-commerce English are significantly affected by macro and micro factors such as South Asian regional culture and target market environments. It is hoped that this study can provide some reference for Chinese enterprises to establish a good brand image in "going to sea".

Due to the privacy of responses to inquiries on most e-commerce platforms, the information pools generally exist in the private chat windows of customer service and customers on e-commerce websites. They will not be open to the public. Visitors who browse the web page can mostly see online consumer reviews. Therefore, in e-commerce discourse analysis, Scholars mainly study online consumer reviews. Online consumer review is a type of product information and work as free "sales assistants" to help consumers identify the products that best match their idiosyncratic usage conditions [4]. Chaudhuri and Ghosh argue that potential users may visualize opinions about specific features of products based on proper analysis and summarization of reviews [5]. Product reviews have always influenced customers more than website information [6]. Investigating this relationship between the company-and-consumer-generated information helps to improve company sales [4, 7]. From the social network perspective, sellers' response to online customer reviews, especially response to negative reviews, has a significant positive effect on their sales [8]. It is necessary for sellers or e-commerce platform companies to be involved in online social conversations. Therefore, proper Interdisciplinary analysis approaches have been put into practice. Sentiment analysis and bidirectional recurrent neural network (RNN) were applied in the analysis of customer reviews [5, 9]. Latent class analysis can better explain the characteristics of the online customer reviews data of durable products for customer segmentation, which may provide support for new product design and development, repositioning of existing products, and product differentiation [10].

The review font may impose a noticeable influence on customer decisions. The feeling of ease in reading led consumers to judge the reviewers as more credible, thus increasing the impact of the reviews. However, the effect of font diminished when there is a high need for cognition, or in an accountable situation [11]. Ban et al. have found top 99 keywords from hotel reviews were divided into four groups such as "Intangible Service", "Physical Environment", "Purpose", and "Location" [12]. A probabilistic language analysis approach has been proposed, in which reviews are translated into engineering characteristics automatically for quality function deployment by estimating the impacts of keywords and nearby words [13].

Besides the lexical research on an online review, there are many papers concerning a grammatical analysis of online communication. Multi-dimensional analysis (MDA) is a standard method to analyze genre variation from the perspective of lexical and grammatical features in 6 dimensions (Dimension 1: Involved & Informational, Dimension 2: Narrative & Non-Narrative, Dimension 3: Context-Independent Discourse & Context-Dependent Discourse, Dimension 4: Overt Expression of Persuasion, Dimension 5: Abstract & Non-Abstract Information, Dimension 6: On-line Informational Elaboration), which is based on the corpus to explain genre features, analyze genre variations [14–17], expand to the analysis of different genre types [18–22], even include the comparison of the translated texts of literary novels [23]. Sun and Cui conducted a metaphor-based multi-dimensional analysis of a self-built business corpus to reveal both semantic features and grammatical features [24]. Previous application of MDA analysis to business and academic English mostly revolves around the discourse of one-way communication, e.g., the English websites of the world's top 500 enterprises with columns

such as mission, vision, and strategy [24], letters to Shareholders [25], academic e-mails [26]. As Skalicky and Crossley figured out that for grammatical function, both *present tense* and *quantification* words can best predict if a text is satirical or non-satirical in the case of Amazon.com product reviews [27].

Different from previous corpora of business or academic English, the *answered questions* on the product page of Daraz (a B2C (business to customer) CBEC platform studied by this paper is a typical genre, which is presented in the form of written text. However, it refers to individual customers rather than enterprise or institutional customers. The language style of customers' questions is either formal or informal; when the cross-border e-commerce platform is oriented to different markets, the language characteristics of customer service replies show salient differences. Customer service staff are sometimes affected by the bilingual context and language style of customers' questions, thereby sometimes adopting a unified formal or informal English language. In recent years, due to the rapid development of the cross-border e-commerce industry, CBEC English, as a new genre of business English, has drawn much attention. Differing from the interdisciplinary research on advertising, enterprise profiles, product titles, and user evaluation of cross-border e-commerce platforms mentioned above, which belong to the one-way transmission of information from enterprises to customers, this paper focuses on the study of the multi-dimensional characteristics of the two-way communication style between enterprises and customers.

## Methodologies

### MDA approach

The MDA approach requires the Multidimensional Analysis Tagger (MAT) that replicates Biber's tagger for the multidimensional functional analysis of English texts [14]. This study adopts Multidimensional Analysis Tagger (v. 1.3) was developed by Nini in 2015, and plotted the input text or corpus on Biber's Dimensions [14]. It determines its closest text type, as proposed by Biber [28]. The study conducts a collection of texts and conversion to machine-readable form, clusters linguistic features into groups of attributes and interpretation of the factors as textual dimensions. Factor scores are regarded to be operational representatives of the textual dimensions.

Then, the research adopts SPSS 22. 0 statistical analysis software to test the significance of the dimension score and Z value of the MAT report, and the dimensions with significant differences ($p < 0.05$) were explained; the variables corresponding to each dimension were stepwise regressed to determine the factors affecting each dimension, and the language features with significant differences were found. And the language features in the two corpora were tested by independent samples t-test, coupled with the most significant 15 features listed sequentially. Finally, the corresponding sub-corpora in the two corpora were further compared and analyzed, particularly the customer questions and replies in different independent country websites compared with the general corpus; and the dimensions and linguistic features with significant differences were analyzed.

### Research corpora

Observed corpus is a Corpus of Answered Questions at Four Country Websites of Daraz in South Asia (CAQF), consisting of four sub-corpora: CAQ-Pakistan, CAQ-Bangladesh, CAQ-Sri Lanka, and CAQ-Nepal, which mainly collects the *answered questions*—a typical interactive communication between customer service staff and the customers in Chinses brands' flagship store in Daraz. Consumer feedback published publicly can influence other consumers' purchase decisions [29]. On the B2C e-commerce platform, customer questions

are equivalent to inquiries, so timely and effective replies directly affect customers' decision-making on purchasing and the subsequent inquiry in the future. This type of communication involves consultation rate and order conversion rate. The process experience of customer consultation and communication is an essential impact factor of the next consultation rate in the store to a certain extent. Excluding other objective factors, whether to get a satisfactory answer after consultation and communication also unavoidably influences on the brand image itself.

Taking a Chinese brand mobile phone as an example, Amazon India, Amazon UK, and other independent stations' "Customer Questions & Answers" are primarily questions and answers between customers, and brand operators or brand parent companies cannot directly participate in the construction of the genre to a large extent. It appears to be relatively passive and is omitted in this study. The same is true of Flipkart and Snapdeal, e-commerce platforms for tens of millions of users in India.

We assembled *answered questions* for the academic research entirely from the flagship stores on four Daraz independent country websites in roughly early 2022, including Daraz Pakistan, Daraz Bangladesh, Daraz Sri Lanka, and Daraz Nepal. Sellers carrying complicated, or high-tech, mass-market products are more likely to benefit from providing consumer reviews than sellers carrying simple, or low-tech, products [4]. Hence, the paper collected *answered questions* related to three categories of available high-tech products, including electronic devices, electronic accessories, TV & Home Appliances from ten Chinese brands: *Gree, Haier, Huawei, Midea, Xiaomi, OPPO, Realme, Redmi, Hisense, Vivo*. A total of 1748 *answered questions* were collected from Daraz Pakistan (461: 19945 tokens in CAQ-Pakistan), Daraz Bangladesh (296: 10904 tokens in CAQ-Bangladesh), Daraz Sri Lanka (474: 19772 tokens in CAQ-Sri Lanka) and Daraz Nepal (517: 19527 tokens in CAQ-Nepal). The language materials from four country websites will be identified as four sub-corpora in part 4. There are many English text materials in Daraz Pakistan, and also in Daraz Sri Lanka. Chinese brands of 3C and major household appliances in the two countries websites rank high and account for a large proportion. There are few language materials that can be collected in Daraz Bangladesh. Even if the products are rated by more than 100 people, there may be no questions at all; in Daraz Bangladesh website, mother tongue is more prevalent because customer service personnel and customers are prone to inconsistent language: customer service personnel in Daraz Bangladesh prefer mother tongue, even if customers ask questions in English, customer service personnel tend to answer in the mother tongue; the proportion of using English in Daraz Nepal website is significantly lower than that of mother tongue, and the choice of language used by customer service staff is consistent with that of customers.

In this paper, reference corpus in the written type is derived from Crown and CLOB with a stratified random sampling of 70 texts each (140 total), which consists of fiction, general prose, learned, and press. The first standard release of Crown and CLOB contains one million words respectively, covering 15 categories in Table 3 of texts published in 2009, or one year before or after 2009. In a few (less than 5%) cases, texts published in 2007 and 2011 were included. CROWN/CLOB corpus of contemporary English is modeled after the sampling strategies of Brown family corpora and established by Chinese scholars. CROWN/CLOB corpus matches its Brown family predecessors (such as Brown, LOB) in size and composition and has been viewed as a good reference corpus for contrastive research of various kinds. The sources of publication come from U.S.-based texts and U.K.-based texts, which were written by the U.S., and the U.K. citizens or permanent residents are selected. For multiple authors, the primary/first author should hold U.S. or U.K. citizenship or permanent residence [30]. Yu and Liang conducted a diachronic two-stage analysis of passive structures with Crown and CLOB as third-generation Brown Family corpus [31]. Lei and Li compared frequency, colligation, and collocation between the corpus of animal science research articles and Crown and CLOB to

**Table 1. Descriptive statistics of corpora.**

| corpora | token | type | STTR(standardized type-token ratio) | Mean word length | mean sentence length | lexical density |
|---|---|---|---|---|---|---|
| CAQF | 70148 | 4664 | 0.65 | 5.48 | 23.13 | 310.14 per 10k |
| Crown & CLOB | 187968 | 20469 | 0.75 | 4.47 | 17.65 | 582.55 per 10k |

develop a deeper understanding of verb usage in animal science research articles [32]. Hence, the sources of selected texts could provide extremely native and precise British English and American English for academic research.

As for the choice of reference, due to historical reasons, British English as the first foreign language and even the official language has already been imposing an important influence on people's daily language in South Asia. Besides, with entry into the fast development era of the internet and e-commerce, U.S. e-commerce platforms and social media, like Amazon, Twitter, and LinkedIn, have been permeating most levels of South Asian society, along with American English. Meanwhile, this study considers that the basic information of household appliances and mobile phone consumers on cross-border e-commerce platforms, such as gender, age, educational background, and occupation, is relatively discrete but not comes from a specific group of population, so it is suitable to use the general language corpus as a reference corpus for research.

By contrast, in Table 1, the observed corpus is remarkably different from the reference corpus in terms of basic statistics, standardized type-token ratio is slightly lower, lexical density is much lower, mean word length is almost the same, and mean sentence is much higher.

This study aims to answer the following questions: 1) What are the different genre features of CBEC English in South Asia? 2) Is there a genre difference among the four sub-corpora? 3) what are the main linguistic features in the corresponding dimensions that generate the genre differences? 4) What are the environmental factors for the occurrence of genre variation?

## Results and discussion

SPSS is used to test the scores of each dimension of CAQF and 140 randomly selected texts from Crown & CLOB English Corpus for independent samples. The ANOVA analysis (Analysis of Variance) was carried out in six dimensions of the four national sub-corpora of the observation corpus and Crown & CLOB. The statistical method of ANOVA applies to the comparison of three or more groups of data, which is also the method used by Biber [15, 16, 33] in his many kinds of literature for the comparison of dimension scores. Then stepwise regression is carried out on the variables corresponding to the dimensions with more significant correlation with the research topics to determine the influencing factors of the two groups of texts in the corresponding dimensions with salient differences.

### Comparative analysis between CAQF and Crown & CLOB

**Independent-sample t-test for each dimension.** The results in Fig 1 & Table 2 of the independent sample t-test of each dimension of CAQF and Crown & CLOB show salient differences between the two corpora in all six dimensions. As shown in Fig 1, both have a score of less than 0 in dimension 1, which belongs to the text with more vital information than involvement, showing important information. Although the texts belong to the question-answer dialogue mode, there is only one pair of question(s) and answer(s) in each group, and there are no more rounds of questions and answers. The purpose of asking questions is to obtain information, so the observed corpus has significant informative characteristics. This research is different from the previous view that business English is more interactive and less informative

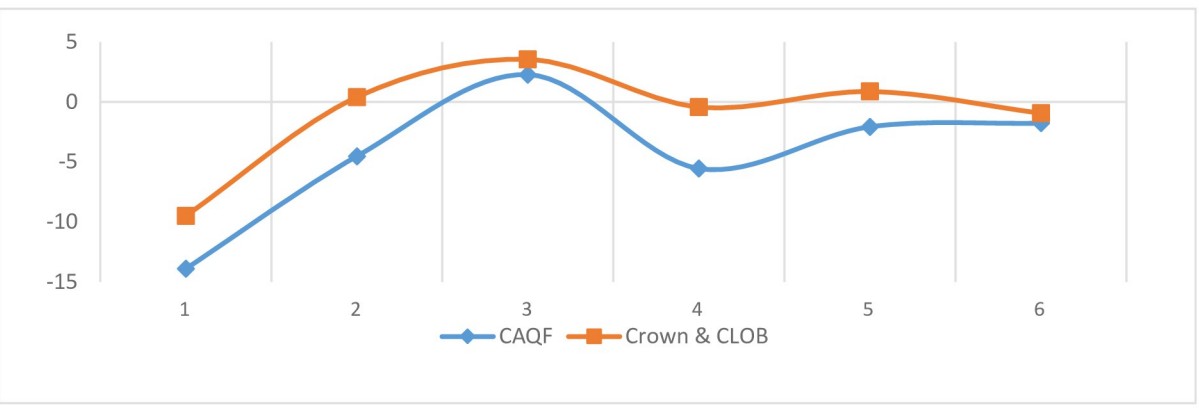

**Fig 1. Dimension differences between CAQF and Crown & CLOB.**

than general English [17], which shows that cross-border e-commerce customer service language pays more attention to the transmission of product and service information and relatively weakens involvement. In the pattern of question-answer, customer questions are regarded as either a request for information or as a request for service [34].

In dimension 2, the observed corpus and reference corpus show opposite dimensional feature. Still the dimension score of CAQF is significantly lower than that of Crown & CLOB, indicating that narrative is not a typical feature of the former, but a feature of general English. The scores in dimension 3 are all positive, showing the characteristics of Context-Independent Discourse, while the Context-Independent attributes of the former are significantly lower than those of the latter. Low scores on this variable indicate that the text of CAQF is relatively dependent on the context.

The dimension scores in dimension 4 are both negative, the reference corpus shows overt persuasion, and the non-persuasion of CAQF is dramatically more potent than that of the reference corpus; Persuasive language tends to exploit language resources to the maximum with catchphrases, emotive words, informal expressions and striking metaphors [35]. Obviously, the text style in CAQF is relatively formal with technical terms and standard phrases. Previous studies suggest that the persuasiveness of general business English is salient and stronger than that of general English [35]. This study draws a different conclusion to confirm the CBEC genre variation.

Both of them show the opposite characteristics, similar to dimension 2 in dimension 5, with a negative score and a positive score, CAQF shows a precise degree of information concreteness, and Crown & CLOB shows a higher degree of information abstraction. Low scores on this variable indicate that the text doesn't provide information in a technical, abstract and

**Table 2. T-test for dimensional differences between CAQF and Crown & CLOB.**

| Significantly Different Dimensions | t | df | Sig. | Mean Difference |
|---|---|---|---|---|
| Dimension1: Involved and Informational | 3.441 | 64.618 | 0.001 | -4.40612 |
| Dimension2: Narrative & Non-Narrative | 7.829 | 168 | 0.000 | -4.93095 |
| Dimension 3: Context-Independent Discourse & Context-Dependent Discourse | 2.330 | 62.860 | 0.023 | -1.27062 |
| Dimension 4: Overt Expression of Persuasion | 8.166 | 168 | 0.000 | -5.12150 |
| Dimension 5: Abstract & Non-Abstract Information | 5.301 | 168 | 0.000 | -2.94076 |
| Dimension 6: On-line Informational Elaboration | 2.902 | 168 | 0.004 | -0.82555 |

formal way. This is consistent with the characteristics of customer service language that both sides often communicate on a specific issue.

In dimension 6, both of them are negative, showing the features of low on-line informational elaboration. CAQF is significantly lower than Crown & CLOB, indicating that the language of customer service personnel responding to customers has not been more finely organized without certain time constraints. Therefore, it is concluded that the CBEC customer service language of the Chinese brands in South Asia belongs to the Learned Exposition genre transmitting specific non-narrative and context-independent information lacking on-line informational elaboration and persuasion, according to a summary of Biber's [28] text types.

**Overall analysis of the factors contributing to the dimensional difference.** Through comparison of the linguistic features in the two corpora by independent sample t-test, it is found that the features with salient differences are as many as 54 (over 80%), which is consistent with the results of comparison [17] between Business English and general English and implicates an absolute genre variation. Table 3 lists the 15 features with the most significant differences.

The learned exposition's features of the observed corpus are apparent. Then observing the language features involved in the multi-dimensional analysis, it is found that NN (Total other nouns), SYNE (Synthetic negation, e.g., *no*, *neither*, *nor*) and POMD (Possibility modals, e.g., *can*, *may*, *might*, *could*) are used more in cross-border e-commerce customer service communication (Z score is greater than 0 and ranks the top three) with more concise language and more specific information. In contrast, the most frequently used in Crown & CLOB are SERE (Sentence relatives, *e.g.*, *Bob likes fried mangoes*, *which is disgusting*), TSUB (That relative clauses on subject position, *e.g.*, *the dog that bit me*), PHC (Phrasal coordination). They tend to be more complex long sentences with distinct levels of sentences and relatively abstract information. The following is a brief illustration of the above views through several examples.

*1. Masum S.: Do you have EMI facility? Is this with 0% interest?*

*Electro Mart LTD: Our EMI service is going on (Only inside the Dhaka City). You can order the product by making online payments from City Bank, Eastern Bank, South East Bank, Lanka Bangla Finance, Mutual Trust Bank, Jamuna Bank or NRB Bank Credit card payment method (In that case no need to call Customer Care and fill the Form). On App, you*

**Table 3. Top 15 features with the largest differences in CAQF and Crown & CLOB.**

| Feature | CAQF | Crown & CLOB | t | Sig. | absolute value of the difference |
|---|---|---|---|---|---|
| NN | 6.74 | 2.16 | 14.692 | 0.000 | 4.58 |
| TSUB | -0.47 | 2.3 | -12.812 | 0.000 | 2.77 |
| [SERE] | 0.22 | 2.8 | -7.142 | 0.000 | 2.58 |
| PHC | -0.28 | 2.18 | -10.225 | 0.000 | 2.46 |
| TO | -1.74 | 0.51 | -11.661 | 0.000 | 2.25 |
| TTR | -1.44 | 0.54 | -7.700 | 0.000 | 1.98 |
| JJ | -1.04 | 0.67 | -11.883 | 0.000 | 1.71 |
| [PRESP] | -0.56 | 0.87 | -13.770 | 0.000 | 1.43 |
| GER | -1.58 | -0.16 | -11.607 | 0.000 | 1.42 |
| PIN | -1.48 | -0.15 | -9.539 | 0.000 | 1.33 |
| PRED | -0.76 | 0.56 | -6.698 | 0.000 | 1.32 |
| ANDC | -0.75 | 0.57 | -14.074 | 0.000 | 1.32 |
| TOBJ | -0.71 | 0.61 | -10.622 | 0.000 | 1.32 |
| [PEAS] | -1.56 | -0.29 | -16.427 | 0.000 | 1.27 |
| [SPAU] | -2.05 | -0.81 | -14.222 | 0.000 | 1.24 |

*can find the help section by clicking the "Account" icon; on Computer, look for "Customer Care" in the top bar. Thank you.*

As shown in the example above, besides the technical term "EMI" in abbreviated form, there are plenty of Total other nouns in the answers, coupled with several Possibility modals "can" and Synthetic negation "*no*".

### Dimensions' ANOVA analysis among sub-corpora and Crown & CLOB

Further, make a comparative analysis among the four sub-corpora of CAQF and Crown & CLOB. The results of the one-way ANOVA test in Fig 2 and Table 4 show that there are salient differences ($p < 0.05$) between the four sub-corpora and Crown & CLOB only in dimensions 2 and 4, and there are no significant differences among the four sub-corpora in all of six dimensions. The reason for the consistency of the genre of the four sub-corpora may be the consistency of regional culture and the same e-commerce genre.

CAQ-Sri Lanka and CAQ-Pakistan differ from the genre of Crown & CLOB, both of which belong to Learned exposition, while the other two national sites, CAQ-Bangladesh and CAQ-Nepal, are closer to the General narrative exposition. In dimension 2, the customer service language of the four national websites shows non-narrative genre feature opposite to that of Crown & CLOB; the ranking of genre differences is CAQ-Sri Lanka, CAQ-Pakistan, CAQ-Nepal, and CAQ-Bangladesh. In dimension 5, only the genre features of CAQ-Pakistan are consistent with the general English, showing a transparent consistency of abstraction; The other three sub-corpora all show a higher degree of specificity of text information, and only in dimension 5, the specificity of customer service language in CAQ-Bangladesh, CAQ-Sri Lanka and CAQ-Nepal remains at a relatively close level, with a slight difference in this dimension. In dimension 1, although both sub-corpora and Crown & CLOB showed obvious informativeness, the former was more informative than the latter, and the sub-corpora with the strongest informativeness and the weakest interactivity were CAQ-Sri Lanka, CAQ-Pakistan,

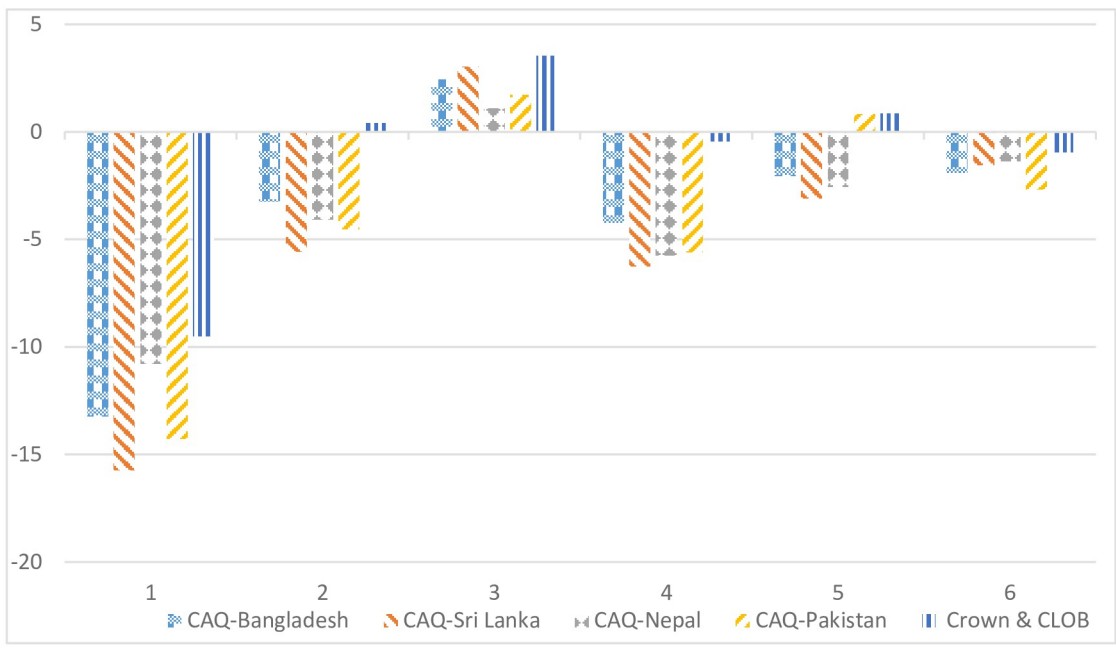

**Fig 2. Dimension differences among four sub-corpora and reference corpus in 6 dimensions.**

**Table 4. The significantly different dimensions.**

| reference corpus | Sub-corpora (observed corpus) | (Dimension 2) p value | (Dimension 4) p value |
|---|---|---|---|
| Crown & CLOB | CAQ-Bangladesh | 0.031 | 0.022 |
| Crown & CLOB | CAQ-Sri Lanka | 0.000 | 0.000 |
| Crown & CLOB | CAQ-Nepal | 0.008 | 0.000 |
| Crown & CLOB | CAQ-Pakistan | 0.008 | 0.004 |

CAQ-Bangladesh, and CAQ-Nepal. In dimension 3, the four sub-corpora and general English all show the characteristics of Context-Independent Discourse, and the three sub-corpora have weaker Context-Independence than the reference corpus. Still there is no salient difference, although the dimension score of CAQ-Nepal seems to be significantly lower than that of reference corpus. In dimension 4, both the observed corpus and the reference corpus have no apparent persuasiveness, and the former is more prominent than the latter. In dimension 6, both the observed corpus and the reference corpus have negative scores on the On-line Informational Elaboration, and CAQ-Pakistan in this dimension has the lowest ranking.

In a word, although CAQ-Sri Lanka and CAQ-Pakistan belong to Learned exposition, the former is more informative, non-narrative, context-independent and persuasive. It possesses more online information elaboration than the latter in dimensions 1, 2, 3, 4, and 6. In dimension 5, the former is more concrete, and the latter is more abstract. Both of the closest text types of CAQ-Bangladesh and CAQ-Nepal are General Narrative Exposition, but in terms of dimensions 1, 3, and 6, the former is more informative and context-dependent referential, with less online information elaboration; In dimensions 2, 4, and 5, the latter is more non-narrative and specific, but less persuasive.

## Regression analysis of the factors affecting the differences within a single dimension

**CAQF.** With SPSS software, the variables corresponding to each dimension were stepwise regressed to determine the influencing factors of the two corpora's texts in six dimensions with salient differences. Due to the limitation of space, only dimension, 1, 3, and 5, which can reflect the stylistic features of the learned exposition, are selected for discussion.

Dimension 1, with 34 variables, is the opposition between Involved and Informational discourse. The regression results showed that 14 variables generating genre variation in dimension 1 entered the regression equation of dimension 1: NN (Total other nouns), PIT (Pronoun *it*), POMD (Possibility modals), VPRT (Present tense), and BEMA (Be as main verb), AWL (Average Word Length), SPP2(Second person pronouns), TTR(Type-token ratio), PRIV(Private verbs), EMPH(Emphatics), FPP1(First person pronouns), SERE(Sentence relatives), COND(Conditional adverbial subordinators) and DPAR(Discourse particles). The adjusted R square value of 0.994 in Table 5 has an excellent predictive effect on dimension 1, which means that the model is more accurate and successful. The standardized regression equation is:

$$\begin{aligned} Dimension\ 1 = & -1.007 - 0.573*NN + 0.952*PIT + 1.197* + POMD + 3.605*VPRT \\ & + 0.886*BEMA - 1.760*AWL + 1.100*SPP2 - 0.857*TTR + 1.241*PRIV \\ & + 1.767*EMPH + 2.238*FPP1 - 0.314*SERE - 0.377*COND \\ & + 3.343*DPAR \end{aligned} \tag{1}$$

**Table 5. Model summary.**

| Model | R | R Square | Adjusted R Square | Std. Error of the Estimate |
|---|---|---|---|---|
| 14 | .999 | .997 | .994 | .42922 |

As shown in Table 6, PIT, Possibility modals, and the other seven variables' coefficients have positive values. The more they are used, the more interactive the text is; Total other nouns, the coefficients of the Average Word Length and the other three variables are negative, and the more they are used, the more informative the text is. For instance, the longer the Average Word Length is, the more the words used in communication are more complex technical terms, such as *installment*, *compartment*, *warranty*, *representative*, etc. The results of independent sample t-test showed that there was salient difference in Average Word Length between CAQF and reference corpus. It is worth mentioning that the linguistic features of Facebook login web service documentation are in line with the textual style of procedural writing, which is mainly informative, clear, direct, and concise with simple present tense, imperative mood, or active voice [36]. However, in dimension 1, the coefficient of Present Tense is positive, which indicates that the smaller the value of this variable, the more informative genre the CAQF is. Other linguistic features, such as Present tense, Discourse particles, First person pronouns [25], etc., showing the speaker's participation in the topic or implying an interactive communication, have lower value since their coefficients seem primarily high.

Dimension 3, with eight variables, is the opposition between Context-Independent Discourse and Context Dependent Discourse. Six variables generating genre variation in dimension 3 entered the regression equation, including WHSUB(WH relative clauses on subject position), NOMZ(Nominalizations), RB(Total adverbs), PHC(Phrasal coordination), TIME (Time adverbials), PLACE(Place adverbials) in Table 7. The adjusted R-square value is equal to 0.985 in Table 8, which has an excellent predictive effect. The standardized regression

**Table 6. Coefficients.**

| Model | | Unstandardized Coefficients | | standardized Coefficients | t | Sig. | Collinearity Statistics | |
|---|---|---|---|---|---|---|---|---|
| | | B | Std. Error | Beta | | | tolerance | VIF |
| 14 | (constant) | -1.007 | .698 | | -1.442 | .170 | | |
| | NN | -.573 | .066 | -.199 | -8.689 | <.001 | .379 | 2.636 |
| | PIT | .952 | .109 | .204 | 8.722 | <.001 | .362 | 2.760 |
| | POMD | 1.197 | .097 | .312 | 12.394 | <.001 | .313 | 3.196 |
| | VPRT | 3.605 | .348 | .242 | 10.358 | <.001 | .365 | 2.742 |
| | BEMA | .886 | .170 | .127 | 5.220 | <.001 | .334 | 2.997 |
| | AWL | -1.760 | .179 | -.211 | -9.855 | <.001 | .434 | 2.306 |
| | SPP2 | 1.100 | .180 | .119 | 6.101 | <.001 | .518 | 1.932 |
| | TTR | -.857 | .089 | -.174 | -9.674 | <.001 | .616 | 1.624 |
| | PRIV | 1.241 | .142 | .154 | 8.730 | <.001 | .641 | 1.559 |
| | EMPH | 1.767 | .305 | .102 | 5.788 | <.001 | .642 | 1.558 |
| | FPP1 | 2.238 | .408 | .098 | 5.481 | <.001 | .618 | 1.619 |
| | SERE | -.314 | .081 | -.069 | -3.881 | .001 | .626 | 1.598 |
| | COND | -.377 | .142 | -.063 | -2.660 | .018 | .358 | 2.793 |
| | DPAR | 3.343 | 1.474 | .046 | 2.269 | .038 | .486 | 2.058 |

**Table 7. Coefficients.**

| Model | | Unstandardized Coefficients | | standardized Coefficients | t | Sig. | Collinearity Statistics | |
|---|---|---|---|---|---|---|---|---|
| | | B | Std. Error | Beta | | | tolerance | VIF |
| 6 | (Constant) | .068 | .389 | | .175 | .863 | | |
| | WHSUB | 2.019 | .172 | .390 | 11.756 | <.001 | .470 | 2.129 |
| | NOMZ | .975 | .064 | .403 | 15.197 | <.001 | .735 | 1.360 |
| | RB | -1.089 | .121 | -.209 | -9.022 | <.001 | .969 | 1.032 |
| | PHC | 1.032 | .081 | .423 | 12.794 | <.001 | .474 | 2.110 |
| | TIME | -.897 | .084 | -.253 | -10.663 | <.001 | .921 | 1.086 |
| | PLACE | -1.041 | .102 | -.260 | -10.224 | <.001 | .803 | 1.245 |

**Table 8. Model summary.**

| Model | R | R Square | Adjusted R Square | Std. Error of the Estimate |
|---|---|---|---|---|
| 6 | .994 | .988 | .985 | .29782 |

equation is:

$$Dimension\ 3 = 0.068 + 2.019 * WHSUB + 0.975 * NOMZ - 1.089 * RB + 1.032 * PHC - 0.897 * TIME - 1.041 * PlACE \quad (2)$$

In Eq (2), the coefficients of three variables—WH relative clauses on subject position, Nominalizations, and Phrasal coordination are positive, which means that the more they are used, the stronger the Context-Independent Discourse nature of the text; For the coefficients of the other three variables, Total adverbs, Time adverbials, and Place adverbials is are negative. More applications of them will make the text present a stronger Context Dependent feature.

Dimension 5, with 8 variables, is the opposition between Abstract and Non-Abstract Information.

Three variables generating genre variation in dimension 5 entered the regression equation, including OSUB (Other adverbial subordinators), CONJ (Conjuncts), and PASS (Agentless passives) in Table 9. The adjusted R-square value is equal to 0.988 in Table 10, and the model is more accurate and successful. Normalized equation: *Dimension 5 = -0.751 + 1.005 * OSUB + 0.957 * CONJ + 0.914 * Pass* (3). It shows that the coefficients of the three variables are all positive, indicating that the fewer Other adverbial subordinators, Conjuncts, and Agentless passives are used, the more obvious the text has Non-Abstract Information features. Mostly, for example, Other adverbial subordinators present the genre's abstractness [17, 20].

**Table 9. Coefficients.**

| Model | | Unstandardized Coefficients | | standardized Coefficients | t | Sig. | Collinearity Statistics | |
|---|---|---|---|---|---|---|---|---|
| | | B | Std. Error | Beta | | | tolerance | VIF |
| 3 | (Constant) | -.751 | .175 | | -4.295 | <.001 | | |
| | OSUB | 1.005 | .035 | .846 | 28.774 | <.001 | .487 | 2.052 |
| | CONJ | .957 | .061 | .325 | 15.624 | <.001 | .977 | 1.024 |
| | PASS | .914 | .159 | .170 | 5.742 | <.001 | .480 | 2.082 |

**Table 10. Model summary.**

| Model | R | R Square | Adjusted R Square | Std. Error of the Estimate |
|---|---|---|---|---|
| 3 | .995 | .989 | .988 | .29037 |

**Sub-corpora.** Next, to briefly illustrate the research path, this paper only develops regression analysis in dimension 2 for an example. Based on the data in Table 4, the results of the regression analysis will be calculated between the observed sub-corpora with the reference corpus. Taking CAQ-Sri Lanka with the most salient difference as an example, the ten independent variables in dimension 2 were subjected to stepwise linear regression analysis. Considering that there was no multicollinearity among the independent variables and that the residual error must obey normal distribution in Table 11, the redundant variables were eliminated. Finally, three independent variables generating genre variation in dimension 2 entered the standard regression equation: SYNE (Synthetic negation), PEAS (Perfect aspect) and PUBV (Public verb) in Table 12. The reference corpus has six independent variables, including VBD (Past tense), PUBV (Public verb), SYNE (Synthetic negation), PRESP (Present participial clauses), PEAS(Perfect aspect), and VPRT(Present tense), shown in Table 12, entering the regression equation in dimension 2, where there are three more variables than the former. According to the standardized coefficient Beta, the top three independent variables in the contribution rate of the two corpora to the dependent variable include SYNE simultaneously. Still, the contribution rate of SYNE to the dependent variable is as high as 85.6% in the observed corpus and only 32.2% in the reference corpus. Meanwhile, the contribution rates of PEAS (Perfect aspect) and PUBV (Public verb) in the two corpora are quite different. Hence, since the three coefficients are all positive, the less use of Synthetic negation, Perfect aspect, and Public verb l will attribute to the features of Non-Narrative in dimension 2, whether expository, descriptive, or other, marked by immediate time and attributive nominal elaboration [28] and mostly present in face-to-face conversation, academic prose, broadcasting, etc. [26, 28].

## Possible reasons for the analysis of environmental factors

This paper explains the reasons for this phenomenon from an interdisciplinary perspective. South Asian CBEC customer service language belongs to the category of business English, which not only presents the use of general CBEC language, but also reflects how customers solve problems in the purchase process through consultation. The author believes that the environmental factors of the genre variation should include the cultural background, market size, Internet infrastructure, and so on of the four observed countries in South Asia.

**Cultural background.** It is assumed that cultural differences affect discourse genres traditionally considered standardized and ritual written business communication [37]. There are some culturally different preferences among countries. In customer service e-mail, Mulken and Meer found that American producers more often express gratitude and Dutch producers are more often sorry to decline a request [2].

According to Hofstede's cultural dimension theory, the four South Asian countries and the United Kingdom and the United States (the reference corpus were collected from the two

**Table 11. Model summary.**

| Model | R | R Square | Adjusted R Square | Std. Error of the Estimate |
|---|---|---|---|---|
| 3 | 1.000 | .999 | .999 | .06992 |
| 6 | .981 | .962 | .960 | .64423 |

**Table 12. Coefficients.**

| Model | | Unstandardized Coefficients | | standardized Coefficients | t | Sig. | Collinearity Statistics | |
|---|---|---|---|---|---|---|---|---|
| | | B | Std. Error | Beta | | | tolerance | VIF |
| 3 | (Constant) | -2.613 | .213 | | -12.285 | <.001 | | |
| | SYNE | 1.007 | .023 | .856 | 44.421 | <.001 | .252 | 3.969 |
| | PEAS | 1.359 | .129 | .105 | 10.531 | <.001 | .936 | 1.069 |
| | PUBV | .988 | .106 | .178 | 9.365 | <.001 | .260 | 3.851 |
| 6 | (Constant) | .309 | .132 | | 2.343 | .021 | | |
| | VBD | 1.941 | .077 | .565 | 25.175 | <.001 | .570 | 1.755 |
| | PUBV | .925 | .052 | .321 | 17.859 | <.001 | .889 | 1.124 |
| | SYNE | .995 | .055 | .322 | 18.176 | <.001 | .913 | 1.095 |
| | PRESP | 1.042 | .047 | .381 | 21.962 | <.001 | .951 | 1.052 |
| | PEAS | .926 | .070 | .240 | 13.277 | <.001 | .880 | 1.136 |
| | VPRT | .541 | .120 | .099 | 4.497 | <.001 | .591 | 1.691 |

countries) present different cultural dimension characteristics. According to Greet Hofstede, two of these four countries, Bangladesh and Pakistan (the research conducted by Hofstede involve neither Nepal or Sri Lanka), get salient different dimension scores in 4 out of the six cultural dimensions from those of the United Kingdom and the United States does severally, including power distance, individualism & collectivism, uncertainty avoidance, long-term oriented & short-term oriented, indulgence & restraint in Table 13. Here are the base culture data for six dimensions of culture as presented in Cultures and Organizations 3rd edition 2010.

Some dimension features match the dimensions of Hofstede's national culture. The differences in dimension 1 and dimension 3 are related to Bangladesh and Pakistan's advocacy of collectivism; the scores are as low as between 14 and 20. Collectivism does not mean closeness. It means that one "knows one's place" in life, which is determined socially. First-person references are related to the way the agent refers to their own personal or corporate identity [38]. In example 2, service staff relied on "our store" instead of the specific brand name, showing a propensity for collectivism.

Dimension 1: a low score on this dimension means that the text presents many nouns, long words, and adjectives (among other features). The data in Tables 1 and 3 could support this point.

"We" or "our" identity:

*2. Q*: The filters are not available in your store.
    *A*: Sir, Available this filter in our store.

**Table 13. Cultural dimension scores.**

| Countries | power distance | Individualism & collectivism | Masculinity & femininity | uncertainty avoidance | long-term oriented & short-term oriented | Indulgence & restraint |
|---|---|---|---|---|---|---|
| Bangladesh | 80 | 20 | 55 | 60 | 47 | 20 |
| Pakistan | 55 | 14 | 50 | 70 | 50 | 0 |
| Great Britain | 35 | 89 | 66 | 35 | 51 | 69 |
| U.S.A. | 40 | 91 | 62 | 46 | 26 | 68 |

Available from: https://geerthofstede.com/research-and-vsm/dimension-data-matrix/

Dimension 3 is also related to collectivism. One is high, and the other is low, which seems to be contradictory. Still in fact, it is the situational dependence of both communication participants that enables strong context communication in this genre.

High-context communication: under this context, communication requires shorter sentences or simple words to deliver information, since many details, like background, norms, and rules, are apparent.

Relatively low scores on this variable indicate that the text depends on the context.

3. *Q*: *Is there any down payment*?
      *A*: *Kindly please contact daraz customer care.*

4. *Q*: *How to do monthly installments*?
      *A*: *You can pay through any bank credit card.*

Dimension 5 is associated with uncertainty avoidance People in the two countries tend to avoid uncertainty (Bangladesh: 60; Pakistan: 70), which is opposite to the situation in developed countries. Bangladeshis are religious, have and maintain an explicit doctrine of faith, adopt a straightforward attitude, and do not tolerate unorthodox ideas and behavior. They are less interested in innovation and pay more attention to correctness, punctuality, and safety factors [39, 40]. Uncertainty avoidance has to do with anxiety and distrust in the face of the unknown, and conversely, with a wish to have fixed habits and rituals, and to know the truth.

Aggression and emotions may sometimes be vented: customers are used to applying several question marks at the end of a question.

5. *Q*: *Installment plans only for credit cards*????

6. *Q*: *Any emi*???

7. *Q*: *Bank price*???

8. *Q*: *Warranty*?????

It is almost easier to find the questions to which the answer may be obvious directly on the home page or product page. Customers choose to get a sure and concrete answer from service staff instead of searching online by themselves.

9. *Q*: *Installment payments available*?
      *A*: *Yes, available.*

10. *Q*: *I need full specification.*
      *A*: *Full description available at the see more option.*

Consumers prefer the official sale of genuine goods. For example, it is very widespread to find conversations around authentic products in smartphone flagship stores:

11. *Q*: *Is this set official*?
      *A*: *Yes.*

12. *Q*: *Is this a fully official phone*??
      *A*: *Dear valued customer, All of our Daraz Mall and Flagship Mobile Phones are Official and Authorized. Thank You.*

13. *Q*: *Is this really a MI Official Flagship store*??
      *A*: *Yes Sir.*

14. *Q*: *Is it launch by India or China*
      *A*: *CHINA.*

**Market scale and internet infrastructure.** With the rapid development of Internet technology, the restrictions on time and space in traditional trade have been eliminated, and cross-border e-commerce, as a new field, has gradually shown its advantages. Cross-border e-commerce is trade networking, based on the Internet. The popularity of the local network, including network speed, cost, policy and other issues, directly affects the development of cross-border e-commerce. Meanwhile, Internet penetration strongly correlates with Hofstede's cultural values [41, 42], thereby exerting influence on linguistic features. The purchasing power of technological innovations, e.g., the Internet [41], mobile phones [43], and B2C e-commerce in high uncertainty-avoidance countries (e.g., Bangladesh and Pakistan) is comparatively weak [44]. This attitude may have an effect on the customers' expectation value of the online replies to their inquiries.

According to 2021–2022 World Bank data in Table 14, among the four countries covered by Daraz, the Internet penetration rate varies from high to low: Nepal, Bangladesh, Pakistan and Sri Lanka. Specially, Sri Lanka has the highest GDP (per capita), but the internet penetration rate maintains the lowest level; whereas, Nepal has the least GDP (per capita), but the internet penetration rate is the highest. The relatively low internet penetration rate tends to restrain the service efficiency and sound development of cross-border e-commerce. Compared with developed countries or regions, where there is a higher level of internet penetration rates of around 90% (European Union: 89%, North America: 94.6%), there would be a promising blue ocean in South Asian CBEC markets.

In CAQ-Sri Lanka, 53.16% of the customer service replies were given one day or even one week after the customer raised the question, while in CAQ-Nepal, the ratio was 43.17%. Among the four Daraz national websites, CAQ-Sri Lanka ranks the highest and CAQ-Nepal does the lowest. Customer response time can reflect the service efficiency of the brand to a certain extent. Although there is not enough evidence to prove that there is an absolute positive correlation between Internet penetration rate and brand service efficiency of cross-border e-commerce platforms, a previous study shows that [9, 45], internet popularization has a significant positive effect on promoting the upgrading of public service consumption, and a higher marginal impact in rural areas. Customers may have been tolerating low-efficiency service responses. Anyhow, the quicker replies with more localized lexicon and grammar in CBEC English genre would undoubtedly impose positive effect on the image of Chinese brands.

15. *Q*: *What is the power Socket type*? *UK American or Australia type*?
   *A*: *US plug.*
   *(** Global Store—answered within 4 days)*

16. *Q*: *My poco m3 dead where can i claim warranty*
   *A*: *Please join live chat so we can share customer service center address*
   *(** Phonezo—answered within 4 days)*

**Table 14. Market scale and internet infrastructure in four countries.**

|  | GDP(per capita) | Internet users | internet penetration rate |
|---|---|---|---|
| Sri Lanka | $3,682(2020) | 7,968,000 (Jun/2021) | 37.10% |
| Pakistan | $1,189 (2020) | 113,679,752 (Jan/2022) | 49.80% |
| Bangladesh | $1,962 (2020) | 117,310,000 (Dec/2021) | 70.10% |
| Nepal | $1,155 (2020) | 21,900,000 (Jan/2022) | 73.00% |

Available from: http://cto.eguidedog.net/node/775

*17. Q*: *will it be available after 14 August???*

 *A*: *Sir restocking date is not yet confirm, please Press the Add to Wishlist button to be notified as soon as it is back in stock*

 *(\*\* Phonezo—answered within 2 days)*

## Conclusions

Besides several basic corpus statistics, including standardized type-token ratio, mean word length, mean sentence length and lexical density, the paper makes a further comparative analysis between the observed corpus CAQF and reference corpus Crown & CLOB by LancsBox. It is concluded that the CBEC customer service language of the Chinese brand in South Asia is classified into the Learned Exposition genre transmitting specific non-narrative and context-independent information lacking on-line informational elaboration and persuasion. The regression results showed that 14 variables, e.g., NN, PIT, POMD, entered the regression equation of dimension 1, six variables, e.g., WHSUB, NOMZ, and RB, entered the regression equation of dimension 3, three variables, including OSUB, CONJ and PASS entered the regression equation of dimension 5. When the coefficients of these variables are positive, the less the variables are used in the texts, the lower the corresponding dimension scores would be and the linguistic features are significant. And the reverse is also true. It is found that the features with salient differences are as many as 54 (over 80%) between the two corpora.

Furthermore, although there isn't a significant genre difference among the four sub-corpora, there are salient differences ($p < 0.05$) separately between the four sub-corpora and Crown & CLOB only in dimensions 2 and 4. Thereinto, taking CAQ-Sri Lanka, which shows the most salient difference with reference corpus as an example, three independent variables, SYNE, PEAS, and PUBV entered the standard regression equation of dimension 2. It is the variable in each dimension equation that determines the linguistic features affects the significant differences between the observed corpora and reference corpus. After tracing back to the original text based on the corpus, the environmental factors for genre variation may be cultural context, South Asian market scale and internet infrastructure.

Halliday put forward the idea of intervening grammar from the language planning perspective [46]. Halliday expanded the ideographic ability of language in specific fields and social activities, so that it can play a new role and promote the evolution of language. Chinese enterprises should reasonably plan the replies to customer questions on the CBEC platform, which can be regarded as a crucial marketing link that can improve the conversion rate, and focus on giving customer answers through concise language to meet customer needs. However, it is not difficult to see that CAQF has the defects of too simple reply, a lack of professional service awareness but with formal linguistic features.

Due to the limitation of research, it is not possible to make a complete regression of all the variables affecting language characteristics in a single dimension in detail; the Hofstede cultural dimension data collected in the analysis does not cover all the countries observed, and only two of them can be used as representatives for the investigation, which is inevitably not rigorous.

From the perspective of building brand image, in addition to *answered questions*, future research can be expanded to the company profile and product description/details that the brand operator or the brand parent company can participate in the production to the greatest extent, as the observation corpus of CBEC English genre variation research. Although these two parts may be considered to be one-way communication from the brand to the user, they play an essential role in building a good brand image.

## Supporting information

**S1 File. Value set of 67 linguistic features or variables.**
(XLSX)

## Acknowledgments

I would like to thank the guest editor and reviewers for their very helpful comments.

## Author Contributions

**Conceptualization:** Shuang Wang.

**Data curation:** Shuang Wang.

**Formal analysis:** Shuang Wang.

**Investigation:** Shuang Wang.

**Methodology:** Shuang Wang.

**Project administration:** Shuang Wang.

**Resources:** Shuang Wang.

**Software:** Shuang Wang.

**Supervision:** Shuang Wang.

**Validation:** Shuang Wang.

**Writing – original draft:** Shuang Wang.

**Writing – review & editing:** Shuang Wang.

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
