## [Decision Letter · Decision Letter 0]

14 Nov 2022

PONE-D-22-23652

A multi-dimensional analysis of CBEC English genre Variation in South Asia

PLOS ONE

Dear Dr. Wang,

Thank you for submitting your manuscript to PLOS ONE. After careful consideration, we feel that it has merit but does not fully meet PLOS ONE’s publication criteria as it currently stands. Therefore, we invite you to submit a revised version of the manuscript that addresses the points raised during the review process.

Dear Author,

Thank you so much for such an interesting paper on CBEC English genre variation in South Asia. It is a data-rich study, which could contribute remarkably to the aspects of business English across the South Asian diaspora. However, the research outcomes lack cohesion with the review of related literature to support the statements and dimensions mentioned by you. The paper requires rearrangement of discussions at different sub-sections to meet the journal’s publication requirements. Our reviewers have shared their valuable inputs for your paper. Please incorporate them accordingly in your paper. Please find the details below in brief.

Reviewer 1:

Add more explanation on why Crown and CLOB is considered as common language corpus in the study. The reasons and explanations are needed, particularly on why Crown and CLOB corpus is chosen as the reference corpus

Reviewer 2:

Please review the entire document for typographical errors, mathematical errors, and any other necessary corrections and etc Following references are missing the Reference mentioned APA Style (Ban, H. J. , Choi, H. , Choi, E. K. , Lee, S. , & Kim, H. S. . (2019), Bickart, B. , & Schindler, R. M. . (2001), Chaudhuri, A. , & Ghosh, S. K. . (2016).,Gong, W. . (2009).Halliday, M. . (1992),Herck, R. V. , Dobbenie, B. , & Decock, S. . (2021),Hu, C.Y., ＆ Tan, J.L.. (2020) ----and so on

Moreover, the paper specifically requires revision on the following aspects for more clarity.

1. The abstract sounds very unclear and just lost in the middle of nowhere. It should be simple and precise enough to describe what the study is about, its methods/tools, and the research outcomes. Its clarity is also required for the purpose of the paper’s alignment with the journal scopes, which does affect its possibility of publication. Modify this please. Also, it is advised to not include any acronym in the abstract.

2. Literature review section requires more elaboration, particularly on literature around MDA and the research methods/tools the study adopts. Moreover, literature related to the mentioned corpora and dimensions need to be reviewed here to support the objectives (RQs in this case). The discussion will also help readers understand the sociocultural communication contexts in the South Asian diaspora.

3. The research corpora (and sub-corpora) and the six dimensions require elaboration with references. This should be either part of the literature review section, or the methodology section after the description of research methods/tools used in the study.

4. The research questions are somewhat confusing. To begin with, basically the study focuses on finding the genre features in CBEC English (RQ 1). RQs 2 and 3 seem repetitive. As dimensions and corpora are mentioned here, it is advised to discuss them just before or after the research questions. To further make these RQs precise, the study can follow this order: RQ 1 about genre features in CBEC English, RQ 2 about the possibility of genre difference among the mentioned corpora, RQ 3 about the identification of linguistic features in the mentioned dimensions affecting the differences if any, and RQ 4 about the identification of environmental factors causing genre variation if any.

5. Results are described at great length. However, this requires thorough discussion and elaboration. Moreover, it misses cohesion in terms of research objectives, reviewed literature and structural construction of paragraphs. For instance, discussion in Section 4.3 is quite sound. This needs to be related to the literature review section to make way for more clarity.

6. It seems readers can make a strong sense of clarity of the topic and related aspects only from the discussion of results. This is why structural reordering is necessary to keep the flow coherent.

7. Conclusion section must not introduce any new aspect or new discussion. It is just a summary of the research.

8. A suggestion here: the paper looks too lengthy with discussions and analyses on genre features, genre differences and linguistic features influencing genre differences. With the suggestions and feedback incorporation accordingly, this will be even lengthier. In such a case, it will be better if you can divide your foci into making two papers: one focusing on the genre features and the other on genre differences along with linguistic and environmental factors affecting it.

9. There are spelling mistakes, sentence construction and grammar issues throughout the manuscript. Please revise it accordingly.

Kindly go through all the comments provided and incorporate them accordingly. Hope to see your revised paper soon.

We look forward to receiving your revised manuscript.

Kind regards,

Dipima Buragohain

Guest Editor

PLOS ONE

Journal Requirements:

3. Please ensure that you refer to Figure 2 in your text as, if accepted, production will need this reference to link the reader to the figure.

4. We note you have included a table to which you do not refer in the text of your manuscript. Please ensure that you refer to Table 5, 7, 11, 14 in your text; if accepted, production will need this reference to link the reader to the Table.

Reviewers' comments:

Reviewer's Responses to Questions

**Comments to the Author**

1. Is the manuscript technically sound, and do the data support the conclusions?

Reviewer #1: Yes

Reviewer #2: Yes

2. Has the statistical analysis been performed appropriately and rigorously? 

Reviewer #1: Yes

Reviewer #2: Yes

3. Have the authors made all data underlying the findings in their manuscript fully available?

Reviewer #1: Yes

Reviewer #2: Yes

4. Is the manuscript presented in an intelligible fashion and written in standard English?

Reviewer #1: Yes

Reviewer #2: Yes

5. Review Comments to the Author

Reviewer #1: 1. Add more explanation on why Crown and CLOB is considered as common language corpus in the study. The reasons and explanations are needed, particularly on why Crown and CLOB corpus is chosen as the reference corpus

Reviewer #2: Please review the entire doccument for typographical erros,mathematical errors, and any other neccessary corrections and etc Following references are missing the Reference mentioned APA Style (Ban, H. J. , Choi, H. , Choi, E. K. , Lee, S. , & Kim, H. S. . (2019), Bickart, B. , & Schindler, R. M. . (2001), Chaudhuri, A. , & Ghosh, S. K. . (2016).,Gong, W. . (2009).Halliday, M. . (1992),Herck, R. V. , Dobbenie, B. , & Decock, S. . (2021),Hu, C.Y., ＆ Tan, J.L.. (2020) ----and so on

6. PLOS authors have the option to publish the peer review history of their article (what does this mean?). If published, this will include your full peer review and any attached files.

Reviewer #1: No

Reviewer #2: **Yes: **MURATHOTI RAJENDRA NATH BABU

---

## [Author Response · Author response to Decision Letter 0]

9 Dec 2022

Dear editor and reviewers,

Thank you for reviewing my manuscript and sending me the letter for revised version.

To reply to questions raised by reviewer 1, I have added more explanation on why Crown and CLOB is considered as common language corpus and the reference corpus.

CROWN/CLOB corpus of contemporary English is modeled after the sampling strategies of Brown family corpora and established by Chinese scholars. CROWN/CLOB corpus matches its Brown family predecessors (such as Brown, LOB) in size and composition on the other and has been viewed as a good reference corpus for contrastive research of various kinds. The sources of publication come from U.S.-based texts and U.K.-based texts which were written by the U.S. and the U.K. citizens or permanent residents are selected. In the case of multiple authors, the primary/first author should hold U.S. or U.K. citizenship or permanent residence. Hence, the sources of selected texts could provide extremely native and precise British English and American English for academic research. 

Due to historical reasons, British English as the first foreign language and even official language has been already imposing an essential influence on people’s daily life in South Asia. Besides, with entry into fast development era of internet and e-commerce, U.S. e-commerce platforms and social media, like Amazon, Twitter, LinkedIn, have been permeating most levels of South Asian society, along with American English. 

To reply to questions raised by reviewer 2, I have made necessary corrections. Meanwhile, all the references mentioned in this paper are listed after the main body. 

Moreover, I finish required revision on 9 aspects for more clarity.

1.Abstract: it has been corrected to be simple and precise enough to describe the 

study purpose, methods and the findings. And the acronym were changed into fully-spelled words.

2. Literature review: 

(1) MDA: I have added more elaboration around six dimensions. However, there are limited literatures around MDA and its essential tool, MAT, a statistical software. In initial submission, most of the literatures I could try my best to search abound MDA were all adopted in this paper. Due to its complexity of statistical approach and metadata privacy, MDA has not been widely applied in linguistic study since 1988. Until Multidimensional Analysis Tagger (v. 1.3) was developed by Nini in 2015, some scholars began to apply MDA to research.

3. I added elaboration with references around reference corpus Crown and CLOB. While observed corpus, including sub-corpora in this research, is a new self-constructed corpus. I added some brief introduction of sub-corpora in specific details to section 3.2. 

4. I rearranged the order of 4 research questions with the logic of “what” first, followed by “why”: 1) What are the different genre features of CBEC English in South Asia? 2) Is there a genre difference among the four sub-corpora? 3) what are the main linguistic features in the corresponding dimensions that generate the genre differences? 4) What are the environmental factors for the occurrence of genre variation? 

5. After reviewing the literatures again, the cohesion has been strengthened.

6. With the logic of research questions and the principle of context consistency, I rearranged the paper structure, especially the structure of Section 4 and added a sub-title 4.4. 

7. I deleted new discussion in Conclusion section.

8. I have to admit the paper sounds a little lengthy due to the further discussion on more qualitative elaboration. However, I believe it is just the combination of genre differences along with linguistic and environmental factors that makes it an innovative and comprehensive research. After all, there has been a criticism aimed at the phenomenon of statistics complicated but linguistics naïve in corpus linguistics. Perhaps, I would delete some sentences to make it shorter and more concise. 

9. I have revised spelling mistakes, sentence construction and grammar errors accordingly.

For Journal Requirements:

1. Manuscript has been corrected to meet PLOS ONE's style requirements, including those for file naming.

2. For my minimal data set, I just provide the value set of 67 linguistic features or variables in observed corpus. Further statistics analysis could be partly conducted based on the set.

3. I arranged Fig1 and 2 as required format both in the text and Figure files separately.

4. I reordered the Tables accordingly to make sure all of them referred in the text.

5. I added three references, No.28, No.38 and No.40, in the list to illustrate precisely the origin and application of Crown and CLOB. And I added No.46 that was missed in the initial manuscript. 

I would like to thank you again for your valuable and significant comments.

I look forward to hearing from you regarding my revision. I would be glad to respond to any further questions and comments that you may have.

Yours

Shuang Wang

---

## [Editor Report · Decision Letter 1]

13 Dec 2022

A multi-dimensional analysis of CBEC English genre Variation in South Asia

PONE-D-22-23652R1

Dear Dr. Wang,

We’re pleased to inform you that your manuscript has been judged scientifically suitable for publication and will be formally accepted for publication once it meets all outstanding technical requirements.

Kind regards,

Guest Editor

PLOS ONE
---

## [Editor Report · Acceptance letter]

4 Jan 2023

PONE-D-22-23652R1 

A multi-dimensional analysis of CBEC English genre variation in South Asia: based on Daraz 

Dear Dr. Wang:

I'm pleased to inform you that your manuscript has been deemed suitable for publication in PLOS ONE. Congratulations! Your manuscript is now with our production department. 

Kind regards, 

on behalf of

Dr. Dipima Buragohain 

Guest Editor

PLOS ONE